# Peel Diffusion and Antifungal Efficacy of Different Fungicides in Pear Fruit: Structure-Diffusion-Activity Relationships

**DOI:** 10.3390/jof8050547

**Published:** 2022-05-23

**Authors:** Gui-Yang Zhu, Ying Chen, Su-Yan Wang, Xin-Chi Shi, Daniela D. Herrera-Balandrano, Victor Polo, Pedro Laborda

**Affiliations:** 1School of Life Sciences, Nantong University, Nantong 226019, China; 2008310007@stmail.ntu.edu.cn (G.-Y.Z.); 1909110087@stmail.ntu.edu.cn (Y.C.); wangsuyan@ntu.edu.cn (S.-Y.W.); shxch0301@ntu.edu.cn (X.-C.S.); daniela.herrera@ntu.edu.cn (D.D.H.-B.); 2Departamento de Química Física, Instituto de Biocomputación y Física de Sistemas Complejos (BIFI), Universidad de Zaragoza, 50009 Zaragoza, Spain

**Keywords:** fungicides, plant fungal pathogens, pear fruit, antifungal activity, peel diffusion, postharvest diseases

## Abstract

Fungal pathogens can invade not only the fruit peel but also the outer part of the fruit mesocarp, limiting the efficacy of fungicides. In this study, the relationships between fungicide structure, diffusion capacity and in vivo efficacy were evaluated for the first time. The diffusion capacity from pear peel to mesocarp of 11 antifungal compounds, including *p*-aminobenzoic acid, carbendazim, difenoconazole, dipicolinic acid, flusilazole, gentamicin, kojic acid, prochloraz, quinolinic acid, thiophanate methyl and thiram was screened. The obtained results indicated that size and especially polarity were negatively correlated with the diffusion capacity. Although some antifungal compounds, such as prochloraz and carbendazim, were completely degraded after a few days in peel and mesocarp, other compounds, such as *p*-aminobenzoic acid and kojic acid, showed high stability. When applying the antifungal compounds at the EC_50_ concentrations, it was observed that the compounds with high diffusion capacity showed higher in vivo antifungal activity against *Alternaria alternata* than compounds with low diffusion capacity. In contrast, there was no relationship between stability and in vivo efficacy. Collectively, the obtained results indicated that the diffusion capacity plays an important role in the efficacy of fungicides for the control of pear fruit diseases.

## 1. Introduction

Pears (*Pyrus* spp.) are one of the most consumed fruits worldwide. The production of pears reached almost 24 million tons in 2019, with main production areas in China, Italy and the USA [1]. The genus *Pyrus* has rich germplasm resources, including thousands of cultivars in five domesticated species and dozens of wild species [2]. Despite its high nutritional and economic relevance [3], pear fruit is highly susceptible to numerous fungal pathogens [4,5]. It has been estimated that fungal diseases account for approximately 20% of perennial yield losses and 10% of postharvest losses [6]. Among pear fungal pathogens, *Alternaria alternata* is a necrotrophic fungus that causes black spot disease [7]. During the early stage of infection, *A. alternata* conidia adhere to the surface of pear fruit, where conidial spores gradually germinate into germ tubes [8]. *A. alternata* releases the host-specific toxins AK-I and AK-II, which block the host defense response [9]. *A. alternata* also secretes cell-wall-degrading enzymes during infection that enable the pathogen to penetrate and infect the host tissue [7,10].

Pear fungal pathogens are difficult to control because they can invade not only the peel but also the external part of the mesocarp and advance through the internal layers of the pear fruit [11]. It was indicated that the pear mesocarp was especially susceptible to *Botrytis cinerea* while the pear was immature [12]. The hydrophobicity of pear fruit cuticular wax is essential in facilitating fungal invasion by regulating the growth and differentiation of *A. alternata* [13,14,15]. Thus, the diffusion capacity of fungicides from peel to the external part of the pear mesocarp may play an important role in their control efficacy.

The control of pear diseases strongly relies on the application of synthetic fungicides [16,17]. Commonly used fungicides can be divided into four categories according to their structural characteristics: (1) Azoles, such as prochloraz and difenoconazole; (2) Benzimidazoles, such as carbendazim; (3) Organosilicons, such as flusilazole; and (4) Thioureas, such as thiram and thiophanate methyl. Although gentamicin is a well-known antibacterial compound, several reports have confirmed that gentamicin is also a powerful antifungal compound [18]. Gentamicin is an aminoglycoside with a high molecular weight (477.60 g/mol). The efficacy of dipicolinic acid (167.12 g/mol) for the management of *Valsa pyri* causing canker disease in pear trees was recently examined [19]. Several natural compounds with low molecular weights (<170 g/mol) and interesting antifungal properties, such as *p*-aminobenzoic acid, kojic acid and quinolinic acid, were reported in the last few years [20,21,22].

Although the effect of diffusion capacity on the in vivo antifungal activity of fungicides has never been studied, several reports have confirmed the ability of fungicides to diffuse from the fruit peel to the mesocarp. For example, Fang et al. reported that prochloraz, pyraclostrobin and tebuconazole diffused in the pear peel and were detected in the pear mesocarp, with tebuconazole having the lowest residue concentration in the pear pulp among the studied fungicides [23]. Pyrimethanil, fludioxonil, cyprodinil and kresoxim-methyl were detected in different layers of apple fruit, and the highest concentrations were found in the apple peel [24]. Shimshoni et al. indicated that the peel penetrability of difenoconazole and tetraconazole in cherry tomatoes was not dependent on the salinity level of the irrigation solution [25]. Utture et al. described that azoxystrobin penetrated the internal parts of pomegranate fruit, while carbendazim and difenoconazole remained in the outer rind [26]. Our research group recently reported that *p*-aminobenzoic acid could penetrate the pear peel and diffuse to the external part of the mesocarp, inhibiting the symptoms of *Colletotrichum fructicola* [27]. Despite some advances, these studies were carried out using only one or a few fungicides with similar structural characteristics.

This study aimed to evaluate the structural characteristics that allow the diffusion of antifungal compounds in pear fruit and how the diffusion capacity influences the antifungal efficacy in vivo. Eleven fungicides with different structural characteristics were used in the study. Diffusion capacity and stability were examined using ‘Cuigan’ pear fruit and the in vivo antifungal activity of the fungicides was screened. *A. alternata*, which is an important pear fruit pathogen and is known to invade the internal phases of the pear fruit, was used in the experiments.

## 2. Materials and Methods

### 2.1. General Information and Strains

Fungicides were used in the experiments as received from commercial suppliers (Table 1). Only technical compounds were used in the screening. *A. alternata* strain HN-5, a Chinese pear pathotype, was used in this study [28]. This fungus was cultured on potato-dextrose-agar (PDA) medium (200 g potato, 20 g dextrose and 15 g agar in 1 L water) at 28 °C.

### 2.2. Measurement of Residue Distribution

Twelve pears were soaked in 100 mL solutions containing 10 mM of different fungicides for 5 min. *p*-Aminobenzoic acid, carbendazim, difenoconazole, dipicolinic acid, flusilazole, gentamicin, kojic acid, prochloraz, quinolinic acid, thiophanate methyl and thiram were used in the screening (Appendix A). This study was designed to evaluate the diffusion capacity and fungicide’s influence on the antifungal activity in pears, hence their concentration was set at 10 mM [19]. To examine the presence of fungicides in pears, one piece of pear (1 cm × 1 cm × 0.4 cm) was extracted using a knife (Appendix A). Each piece was carefully divided into peel and mesocarp [27]. Each sample was placed in an Eppendorf tube with 100 μL methanol, and the resulting suspension was vortexed for 5 min. The presence of fungicides was analyzed using a high-performance liquid chromatography (HPLC) system (Agilent 1200 Series, USA) equipped with a C18 column (Eclipse XDB-C18 column, 250 × 4.6 mm, Agilent, Santa Clara, CA, USA) and UV detector. Detailed separation conditions are indicated in Table 2 [14,27,29,30]. Experiments were repeated five times.

In the case of gentamicin, the extracted samples were placed in Eppendorf tubes and 100 μL water was added. The resulting suspension was vortexed for 5 min. Fifty microliters of the suspension were mixed with 20 μL aqueous sodium bicarbonate (1 M) and 50 μL acetone containing Marfey’s reagent (0.1 M) [30]. The reaction solution was stirred at 180 rpm and 37 °C for 3 h.

Pesticide diffusion level (PDL) was calculated using the following formula: PDL (%) = (X/Y) × 100; where ‘X’ is the peak area of pesticide in the pear mesocarp and ‘Y’ is the sum of the peak areas in peel and mesocarp [31]. All results were pooled to calculate the mean value and deviation.

### 2.3. Computational Studies

Dipole moments and molecular volumes were obtained from electronic structure calculations at the density functional theory (DFT) level carried out using the Gaussian program package [32]. The M06-2X exchange-correlation functional [33] combined with the 6-311G(d,p) basis set and solvent corrections using the solvation model based on density (SMD) method [34] for water were selected. The nature of the stationary points was confirmed by analytical frequency analysis. Molecular volumes were calculated using a density isocontour value of 0.001 electrons/Bohr^3^.

### 2.4. Stability Assay

Pear fruit (*Pyrus pyrifolia* ‘Cuigan’) were treated with fungicides following the conditions reported in Section 2.2. Twelve pears were used for the screening of each fungicide. The samples were collected after 0, 1, 2, 3, 5 and 10 days. Twelve samples (one sample was extracted from each pear) were used for each time point. The concentrations of fungicides were calculated according to the peak area. Standard curves were established for each fungicide from 0 to 10 mM. Experiments were repeated five times.

### 2.5. In Vitro Antifungal Assay

The antifungal activities of the eleven fungicides were examined using *A. alternata* HN-5. Briefly, the fungal pathogen was placed in the center of a Petri dish containing PDA medium and 0.1, 0.2, 0.5, 1, 2, 5, 10, 20, 30, 40 and 50 mM of each fungicide [35]. The negative control experiment was performed by culturing *A. alternata* on PDA in the absence of fungicides. The dishes were incubated at 28 °C for 3 days. Experiments were repeated five times. Half maximal effective concentration (EC_50_) refers to the concentration of fungicide that can reduce mycelial growth by 50%. EC_50_ values were calculated by the least-squares method using Prism 7.0 (GraphPad Software, San Diego, CA, USA) [36,37].

### 2.6. In Vivo Antifungal Assay

The antifungal efficacies for the control of *A. alternata* HN-5 in pear fruit (*P. pyrifolia* ‘Cuigan’) were screened by applying the fungicides at the EC_50_ concentrations. *A. alternata* was cultured on PDA medium at 28 °C for five days, and the mycelium was divided into 7-mm-diameter plugs. After soaking the pears in 100 mL aqueous solutions containing the fungicides (in the EC_50_ concentration) for 30 min, the pears were dried at room temperature and 5-mm-diameter wounds were made on the surface of the pear using a sterilized knife. Sterile water, in the absence of fungicides, was used for the control experiment. Then, a mycelial plug was kept in contact with the wound. The efficacy was measured according to the lesion length caused by *A. alternata* after 3, 4, 5 and 6 days of incubation at 28 °C and 70% humidity. The antifungal efficacy was calculated according to the diameter of the lesion length using the following formula: antifungal efficacy = [1 − T/C] × 100, where T is the diameter of the lesion length for the treatment sets, and C is the diameter of the lesion length for the control sets [38]. *A. alternata* caused brown lesions on the fruit peel around the inoculation site that were observed with the naked eye. Twenty-five pears were used for each treatment condition. Experiments were repeated three times.

### 2.7. Statistical Analysis

The obtained data were analyzed by the one-way ANOVA method and the significance levels were calculated by Tukey’s multiple range test. Differences between means were considered significant when *p* ≤ 0.05. ANOVA and linear regression analyses were performed using the SPSS software (version 16.0).

## 3. Results and Discussion

### 3.1. Correlations between Peel Diffusion and Structural Characteristics of Fungicides

Eleven fungicides, including *p*-aminobenzoic acid, carbendazim, difenoconazole, dipicolinic acid, flusilazole, gentamicin, kojic acid, prochloraz, quinolinic acid, thiophanate methyl and thiram were used in this study. Carbendazim, difenoconazole, flusilazole, prochloraz, thiophanate methyl and thiram have been commercialized for agricultural uses. The antifungal properties of *p*-aminobenzoic acid, dipicolinic acid, gentamicin, kojic acid and quinolinic acid against some plant pathogens have been examined; however, these compounds are only experimental and have never been commercialized for agricultural use. When selecting molecules for the purpose of this study, compounds with a variety of structural characteristics were prioritized to show how these structural differences influence diffusion capacity, stability and, thus, in vivo efficacy. The effects of the structural characteristics of fungicides on their in vitro antifungal activities have been thoroughly studied, and several structure/activity relationship studies can be found in the literature [39,40]. However, there is a lack of information regarding the effect of the structural characteristics on the in vivo activity and how the fungicides behave in the host plants [41].

As can be seen in Table 3, the highest diffusion capacities were observed for *p*-aminobenzoic acid, carbendazim and quinolinic acid (PDL = 0.205, 0.138 and 0.177, respectively). It must be noted that these compounds showed some of the lowest volumes (91.532, 135.058 and 113.031 cm^3^/mol, respectively) among all studied compounds, suggesting that small compounds show higher diffusion capacities in pear peel compared to large compounds. Accordingly, gentamicin and flusilazole, which have large volumes (339.397 and 214.571 cm^3^/mol, respectively), were only detected in the pear peel but not in the pear mesocarp, indicating that they are unable to penetrate the peel to reach the internal phases of the pear fruit. The minimum energy conformations of the studied fungicides are shown in Figure 1A–K, and energy and cartesian coordinates are shown in Appendix A.

When comparing compounds with small volumes, it can be observed that other factors affect diffusion. For example, *p*-aminobenzoic acid and quinolinic acid showed high diffusion capacities (PDL = 0.205 and 0.177, respectively), whereas the diffusion capacities of dipicolinic acid and kojic acid were much lower (PDL = 0.029 and 0.023, respectively). As shown in Table 3, the dipole debyes of *p*-aminobenzoic acid and quinolinic acid (2.0528 and 2.5696, respectively) were lower than those calculated for dipicolinic acid and kojic acid (7.7607 and 7.233, respectively), suggesting that the polarity of the compounds also affects the diffusion capacity, with nonpolar compounds showing higher diffusion capacities than polar compounds.

A linear regression analysis was used to confirm the relationship between polarity and size and diffusion capacity. The obtained results showed that the influence of polarity and size on the diffusion capacity was 58.6%, with both polarity (*p* < 0.001) and size (*p* < 0.001) having significant effects and being negatively correlated with the diffusion capacity (Figure 2A). The calculated regression equation considering polarity and size was: diffusion capacity = 0.212 − 0.001 × volume − 0.011 × dipole debye; suggesting that, although both polarity and size are involved in the diffusion capacity, the former seems to have greater effect than size.

Although no similar study has been published to date, there are several reports using fruit peels to adsorb organic compounds. In this field, dragon fruit, grape, mango and persimmon peels have been used for the adsorption of organic dyes in water [42,43,44,45]. The primary layer of the pear skin is the cuticle, which is formed by polyester cutin and a mixture of lipidic compounds, and is highly hydrophobic [46,47]. In addition, pear peel contains up to 16% of lignin, a nonpolar polymer formed by cross-linked phenolics that shows affinity toward nonpolar aromatic structures via the formation of π interactions [48,49]. The presence of these structures in the pear skin may explain the higher diffusion of nonpolar fungicides compared to polar fungicides. However, further studies are necessary to understand the molecular interactions that are involved in the diffusion effect. In agreement with our results, the diffusion of organic toxins in pear leaves was reported to be negatively correlated with the molecular weight of the compounds, with large compounds showing low diffusion rates [50,51]. However, it was indicated that the diffusion in leaves was not related to polarity.

As indicated above, both polarity and size influenced the diffusion capacity in pear fruit by 58.6%, showing that there are other factors that affect the diffusion capacities of the compounds. Chan et al. reported that, for molecules of the same size, planar solutes were able to diffuse faster than spherical solutes [52]. Although the shape may be an important factor influencing the diffusion capacity in pear peels, only volume and polarity were considered in this study.

### 3.2. Stability of Fungicides in Pear Peel and Mesocarp

The stability of the eleven fungicides in both pear peel and mesocarp is shown in Figure 3A–K. The concentrations of most fungicides gradually decreased over time, indicating their degradation. In this sense, the peak corresponding to prochloraz completely disappeared in the fruit peel after three days, and in the pear mesocarp after two days. Similarly, quinolinic acid was completely degraded in both peel and mesocarp after five days, while the complete degradation of gentamicin in pear skin was observed after three days. Carbendazim, dipicolinic acid, thiophanate methyl and thiram were detected in the pear peel for 10 days, with 94%, 99%, 84% and 58% degradation rates, respectively; while none of the compounds were observed in the mesocarp after 10 days. These results indicated that prochloraz, quinolinic acid and gentamicin can protect pear fruit during short periods, limiting their possible application for the control of pear diseases. Similarly, carbendazim, dipicolinic acid, thiophanate methyl and thiram degraded rapidly in both skin and mesocarp, suggesting that these compounds are not suitable for the protection of pear fruit during long periods. In contrast with our result, prochloraz was reported to show high stability in skin and mesocarp of ‘Dangshan Su’ pears [23], which suggests that the pear variety plays a key role in the stability of fungicides. Zhao et al. indicated that prochloraz showed high stability in bayberries [53].

*p*-Aminobenzoic acid, difenoconazole and kojic acid were detected in both peel and mesocarp for 10 days. These results demonstrated that *p*-aminobenzoic acid, difenoconazole and kojic acid can be used for pear mesocarp protection during reasonable periods. Although flusilazole was not detected in the mesocarp, this fungicide showed high stability in pear peel. Kojic acid showed one of the highest stabilities among the studied fungicides, with degradation rates of 34% and 50% in the peel and mesocarp, respectively, after 10 days, while the degradation of *p*-aminobenzoic acid was 86% and 78% in peel and mesocarp, respectively, after 10 days. It must be noted that the concentration of difenoconazole remained unchanged in both skin and mesocarp, indicating that this fungicide shows high stability in pear fruit.

Thioureas thiophanate methyl and thiram degraded rapidly in the mesocarp, but they showed good stability in the pear peel. In contrast, the degradation of *p*-aminobenzoic acid was faster in the pear peel than in the pear mesocarp. In agreement with these results, several studies have indicated that the degradation rates are highly dependent on the fungicide and plant tissue. For example, Dong et al. reported that triazole alcohol fungicides had different degradation rates in tissues of cucumber fruit [54]. Li et al. indicated that the dissipation rate of imazalil varied in different plant tissues [55]. The stability of dipicolinic acid was previously studied in pear bark, showing no degradation during 20 days [19]; however, dipicolinic acid was completely degraded after 10 days in the pear mesocarp.

Linear regression analyses indicated that neither volume (*p* > 0.05) nor polarity (*p* > 0.05) were significantly correlated to the stability of the compounds (Appendix A).

### 3.3. Efficacy of Fungicides for the Control of A. alternata in Pear Fruit

To determine the effect of diffusion and stability on the antifungal activity, the same in vitro inhibitory rate was used in the in vivo experiments by calculating the EC_50_ values. The lowest EC_50_ value was detected for prochloraz (EC_50_ = 0.13 mM), followed by flusilazole (EC_50_ = 0.51 mM), thiram (EC_50_ = 0.52 mM), difenoconazole (EC_50_ = 0.79 mM), dipicolinic acid (EC_50_ = 4.5 mM), gentamicin (EC_50_ = 5.3 mM), thiophanate methyl (EC_50_ = 11.1 mM) and *p*-aminobenzoic acid (EC_50_ = 11.4 mM). On the other hand, carbendazim (EC_50_ = 20.1 mM), kojic acid (EC_50_ = 20.3 mM) and quinolinic acid (EC_50_ = 46.9 mM) showed the highest EC_50_ values (Table 4). The fungicides flusilazole, prochloraz and thiram at 5 mM concentration each completely inhibited *A. alternata* mycelial growth, indicating strong antifungal activity against the pathogen. In contrast, the mycelial growth of *A. alternata* was not significantly altered by treatment with 5 mM quinolinic acid.

In agreement with the EC_50_ values obtained in this study, Nallathambi et al. indicated that thiophanate methyl showed weak antifungal efficacy for the management of *A. alternata* on postharvest *Ziziphus mauritiana* [56]. Several reports have indicated that carbendazim showed weak activity to *A. alternata*, with similar inhibitory values as the one reported here [56,57]. The EC_50_ values obtained for carbendazim and thiophanate methyl indicated that *A. alternata* strain HN-5 shows resistance to these fungicides. Numerous *A. alternata* strains with high levels of resistance to commonly used fungicides were reported recently [58,59]. In agreement with the strong antifungal activity of prochloraz against *A. alternata* HN-5, this fungicide was also reported to efficiently inhibit the symptoms of *A. alternata* in mango and persimmon [60]. The EC_50_ values of difenoconazole to *A. alternata* C02, XY29, XY36 and XY45 ranged from 0.73 to 1.13 mM [61], which are consistent with the EC_50_ value obtained for difenoconazole in this work (EC_50_ = 0.79 mM). Although Vujanovic et al. reported that 4.16 mM thiram only inhibited the mycelial growth of an *A. alternata* (using a strain isolated from asparagus) by 25% [62], the EC_50_ of thiram to *A. alternata* HN-5 was 0.52 mM, indicating that HN-5 is more sensitive than the other strain. As far as we know, this is the first study on the antifungal activities of dipicolinic acid, flusilazole, gentamicin, *p*-aminobenzoic acid, kojic acid and quinolinic acid to *A. alternata*.

Regarding the in vivo assay, *p*-aminobenzoic acid, carbendazim and quinolinic acid strongly inhibited *A. alternata* symptoms (Figure 4 and Table 5). For example, *p*-aminobenzoic acid reduced *A. alternata*-caused lesions by 43%, 44%, 50% and 46% after 3, 4, 5 and 6 days, respectively; while carbendazim inhibited the lesion length by 28%, 45%, 42% and 38% after 3, 4, 5 and 6 days, respectively. It must be noted that *p*-aminobenzoic acid, carbendazim and quinolinic acid showed high diffusion capacity, suggesting that diffusion capacity is a determining factor in the in vivo antifungal activity.

In contrast, dipicolinic acid, kojic acid, prochloraz and thiram, which showed low diffusion capacity, did not efficiently control the advancement of the pathogen. One of the lowest efficacies was observed for kojic acid, which only inhibited the lesion length by 0%, 2%, 1% and 0% after 3, 4, 5, and 6 days, respectively. Similarly, the lesion length caused by *A. alternata* after treatment with difenoconazole and flusilazole did not change compared to the control experiment. As indicated in the “Correlations between peel diffusion and structural characteristics of fungicides” and “Stability of fungicides in pear peel and mesocarp” sections, difenoconazole, flusilazole and kojic acid showed low diffusion capacities and high stability.

The high efficacy of the fungicides with high diffusion capacity can be explained considering that these compounds are present in the pear mesocarp and, thus, protect the internal parts of the pear fruit, limiting the pathogen advancement. All fungicides were applied using the same in vitro antifungal activity and, thus, the intrinsic antifungal activity of the fungicides was not considered. The main objective of this study was to examine the effect of the diffusion capacity of the fungicides on the in vivo antifungal activity. For this reason, the suggested commercial rates were not followed.

To confirm the correlations between diffusion/stability and in vivo antifungal activity, a linear regression analysis was carried out. The obtained results showed that diffusion influenced the in vivo activity by 66.2% (*p* < 0.001); however, the stabilities in peel (*p* > 0.05) and mesocarp (*p* > 0.05) did not significantly affect the antifungal efficacy. The regression equation was: antifungal efficacy = −0.001 + 1.941 × diffusion capacity (Figure 2B).

Although fungicides with high diffusion capacity allowed the control of *Alternaria alternata* in pear, it must be noted that the presence of toxic fungicides in the mesocarp poses a food safety hazard. Although the removal of the fungicides from the fruit peel can be done easily by peeling with a knife, the removal of the fungicides from the mesocarp seems a difficult task. Some of the studied fungicides are known to cause negative effects on human health. For example, carbendazim has been related to cancer, and is known to produce infertility, hepatocellular dysfunction and developmental toxicity in mammals [63]. Difenoconazole, flusilazole and prochloraz are known to produce developmental toxicity and to cause malformation during embryo development [64,65,66], while thiophanate methyl and thiram have been reported to show hepatotoxic effects [67,68].

Our results suggested that the stability was not correlated with the in vivo antifungal activity; however, the inoculation of *A. alternata* was carried out just after the treatment. The stability must be involved in the in vivo efficacy when considering different inoculation time points, and the stable compounds may be more efficient than the unstable compounds for long periods.

## 4. Conclusions

In summary, this study has revealed for the first time that size and polarity play an important role in the diffusion capacity of fungicides in pear fruit, with small and non-polar fungicides showing higher diffusion capacity than those with large size and high polarity. At the same time, the diffusion capacity was found to be positively correlated with the in vivo antifungal activity of the fungicides. This study provides new insights regarding the structural characteristics that are suitable for the control of fungal pathogens in pear fruit and will lead to a better understanding of the factors involved in the in vivo efficacy of fungicides.

## Figures and Tables

**Figure 1 jof-08-00547-f001:**
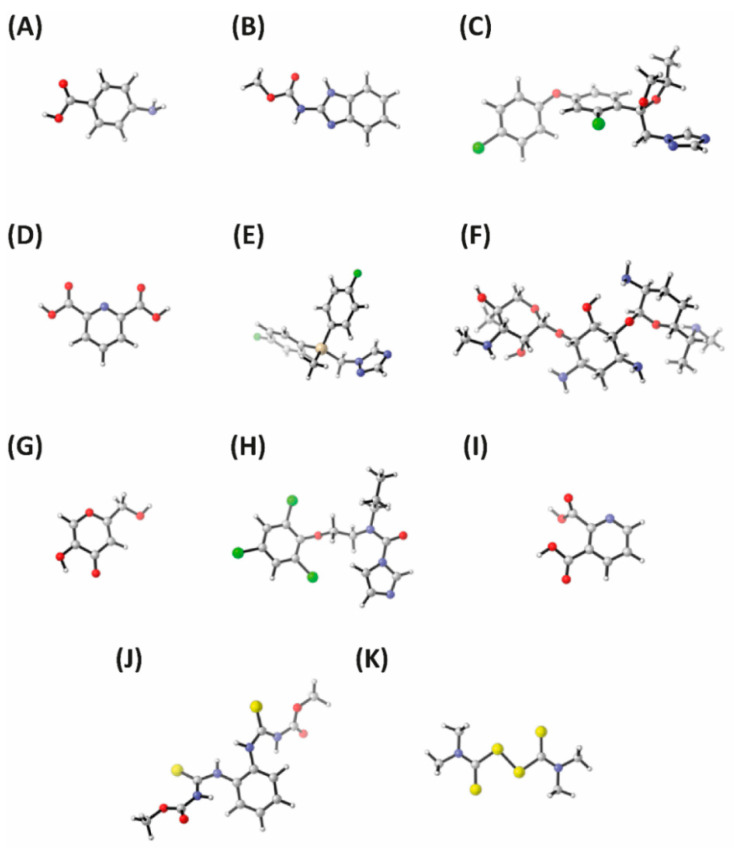
Density functional theory (DFT; M06-2X(PCM)/6-311G(d,p)) optimized structures for the studied fungicides. (**A**) *p*-Aminobenzoic acid. (**B**) Carbendazim. (**C**) Difenoconazole. (**D**) Dipicolinic acid. (**E**) Flusilazole. (**F**) Gentamicin. (**G**) Kojic acid. (**H**) Prochloraz. (**I**) Quinolinic acid. (**J**) Thiophanate methyl. (**K**) Thiram. All molecules are shown at the same scale to allow size comparisons.

**Figure 2 jof-08-00547-f002:**
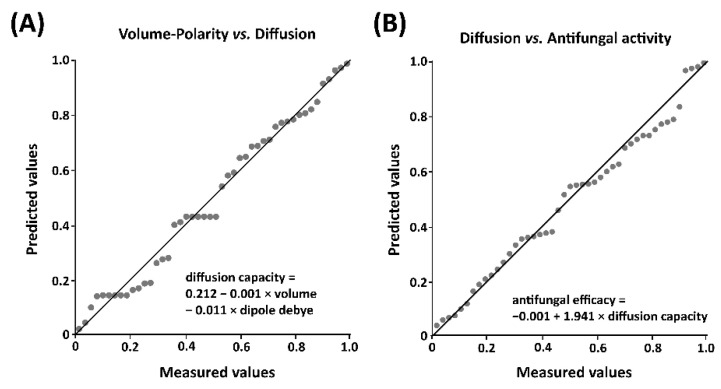
Linear regression analyses. (**A**) Correlation between volume, polarity and diffusion. (**B**) Correlation between diffusion and in vivo antifungal activity. Linear regression analyses were performed using SPSS software version 16.0.

**Figure 3 jof-08-00547-f003:**
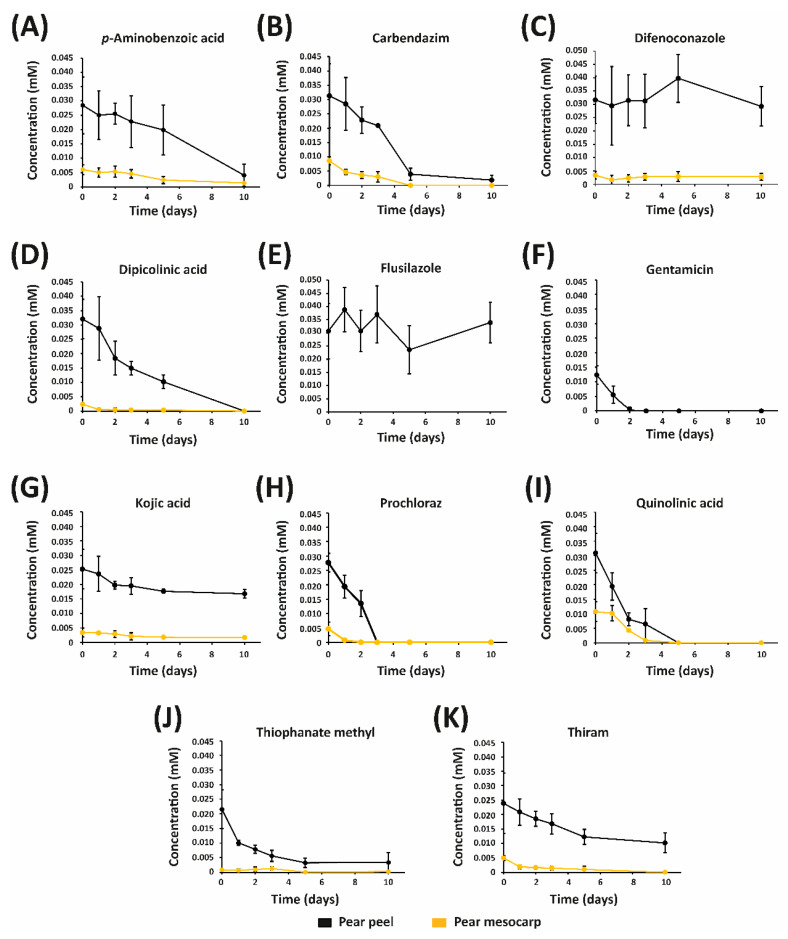
Stability of studied fungicides in pear peel and mesocarp. (**A**) *p*-Aminobenzoic acid. (**B**) Carbendazim. (**C**) Difenoconazole. (**D**) Dipicolinic acid. (**E**) Flusilazole. (**F**) Gentamicin. (**G**) Kojic acid. (**H**) Prochloraz. (**I**) Quinolinic acid. (**J**) Thiophanate methyl. (**K**) Thiram.

**Figure 4 jof-08-00547-f004:**
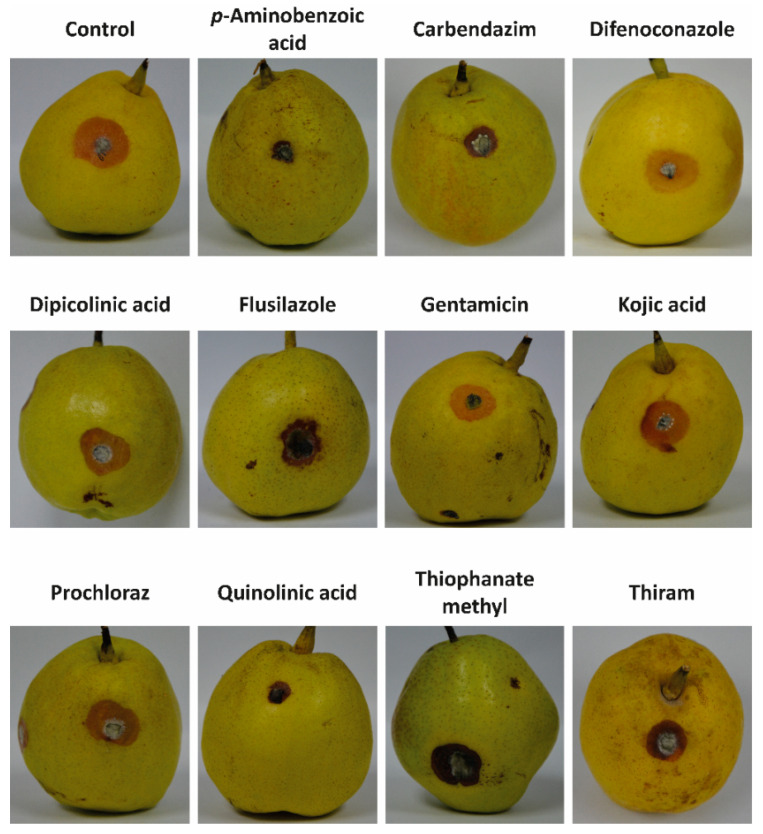
Images of the symptoms caused by *Alternaria alternata* on pear fruit after application of *p*-aminobenzoic acid, carbendazim, difenoconazole, dipicolinic acid, flusilazole, gentamicin, kojic acid, prochloraz, quinolinic acid, thiophanate methyl and thiram. The fungicides were applied at the EC_50_ concentration. The control experiment was carried out in the absence of fungicides.

**Table 1 jof-08-00547-t001:** Information of the antifungal compounds used in this work.

Standard Name	Chemical Name	Molecular Weight (g/mol)	Commercial Source
*p*-Aminobenzoic acid	*p*-Aminobenzoic acid	137.14	Macklin (Shanghai, China)
Carbendazim	Methyl benzimidazole-2-ylcarbamate	191.19	Shyuanye (Shanghai, China)
Difenoconazole	1-[2-[2-Chloro-4-(4-chloro-phenoxy)-phenyl]-4-methyl[1,3]dioxolan-2-ylmethyl]-1H-1,2,4-triazole	406.26	Aladdin (Shanghai, China)
Dipicolinic acid	2,6-Pyridinedicarboxylic acid	167.12	Macklin
Flusilazole	Di(4-fluorophenyl)(1,2,4-triazole-2-ylmethyl)methylsilane	315.39	Macklin
Gentamicin	2-[4,6-Diamino-3-[5-amino-2-[1-(methylamino)ethyl]oxan-4-yl]oxy-2-hydroxycyclohexyl]oxyoxan-3-ol	477.60	Aladdin
Kojic acid	5-Hydroxy-2-hydroxymethylgamma-pyrone	142.11	Macklin
Prochloraz	*N*-Propyl-*N*-[2-(2,4,6-trichlorophenoxy)ethyl]imidazole-1-carboxamid	376.67	Aladdin
Quinolinic acid	2,3-Pyridinedicarboxylic acid	167.12	Macklin
Thiophanate methyl	Dimethyl(1,2-phenylene)*bis*(iminocarbonothioyl)*bis*(carbamate)	342.39	Aladdin
Thiram	Tetramethylthiuram disulfide	240.43	Macklin

**Table 2 jof-08-00547-t002:** High-performance liquid chromatography (HPLC) conditions used for the detection of fungicides.

Fungicide	HPLC Method	Retention Time (min)	Absorption Wavelength (nm)
*p*-Aminobenzoic acid	^1^	5.6	283
Carbendazim	^2^	20.1	254
Difenoconazole	^2^	20.3	254
Dipicolinic acid	^3^	6.7	270
Flusilazole	^2^	17.3	254
Gentamicin	^4^	35.7	350 ^5^
Kojic acid	^3^	6.1	280
Prochloraz	^2^	19.2	254
Quinolinic acid	^3^	7.5	270
Thiophanate methyl	^2^	9.2	254
Thiram	^2^	12.1	254

^1^ HPLC conditions: 1 mL/min flow (column temperature: 25 °C; injection volume: 10 μL). The mobile phase was 20% to 45% CH_3_CN in H_2_O from 0 to 5 min, 45% to 50% CH_3_CN in H_2_O from 5 to 19 min, 50% to 60% CH_3_CN in H_2_O from 19 to 20 min, 60% to 100% CH_3_CN in H_2_O from 20 to 23 min, 100% CH_3_CN from 23 to 27 min, 100% to 20% CH_3_CN in H_2_O from 27 to 28 min, and 20% CH_3_CN in H_2_O from 28 to 30 min (H_2_O contained 0.04% trifluoroacetic acid) [27]. ^2^ HPLC conditions: 1 mL/min flow (column temperature: 25 °C; injection volume: 5 μL). The mobile phase was 30% to 100% CH_3_CN in H_2_O from 0 to 30 min, and 30% CH_3_CN in H_2_O from 30 to 35 min [29]. ^3^ HPLC conditions: constant flow, 0.3 mL/min, 0.03 M H_2_SO_4_ aqueous solution (column temperature: 60 °C; injection volume: 10 μL) [19]. ^4^ HPLC conditions: 1 mL/min flow (column temperature: 25 °C; injection volume: 50 μL). The mobile phase was 0% to 60% CH_3_CN in H_2_O from 0 to 60 min, 90% CH_3_CN in H_2_O from 60 to 70 min, and 0% CH_3_CN in H_2_O from 70 to 75 min [30]. ^5^ After labeling with Marfey’s reagent [30].

**Table 3 jof-08-00547-t003:** Diffusion capacity and structural characteristics of the studied fungicides.

Fungicide	Diffusion Ability ^1^	Volume ^2^ (cm^3^/mol)	Dipole ^2^ (debye)
*p*-Aminobenzoic acid	0.205 ± 0.055 a	91.532	2.0528
Carbendazim	0.138 ± 0.036 ab	135.058	4.3805
Difenoconazole	0.055 ± 0.032 cd	285.34	6.0014
Dipicolinic acid	0.029 ± 0.017 cd	111.23	7.7607
Flusilazole	0 e	214.571	3.6535
Gentamicin	0 e	339.397	3.1864
Kojic acid	0.023 ± 0.006 d	93.169	7.233
Prochloraz	0.057 ± 0.014 c	240.481	6.6457
Quinolinic acid	0.177 ± 0.020 a	113.031	2.5696
Thiophanate methyl	0.047 ± 0.019 cd	248.402	6.7427
Thiram	0.094 ± 0.049 bc	153.518	0.0088

^1^ Differences between means were considered significant when *p* ≤ 0.05. ^2^ Volume and dipole debye were calculated using DFT methodology (M06-2X(PCM)/6-311G(d,p)). Minimum energy conformations are shown in Figure 1.

**Table 4 jof-08-00547-t004:** In vitro antifungal activities of studied fungicides against *Alternaria alternata*.

Fungicide	Mycelial Growth ^2,3^ (mm)		EC_50_ (mM)
	1 mM Fungicide	5 mM Fungicide	
*p*-Aminobenzoic acid	32.2 ± 1.6 a	25.3 ± 1.4 c	11.4 ± 0.3 c
Carbendazim	34.0 ± 0.7 a	30.1 ± 0.9 b	20.1 ± 0.4 b
Difenoconazole	15.3 ± 1.0 c	9.0 ± 0.9 e	0.79 ± 0.11 f
Dipicolinic acid	29.8 ± 0.7	18.2 ± 1.3 d	4.5 ± 0.2 e
Flusilazole	6.9 ± 0.2 e	-	0.51 ± 0.06 f
Gentamicin	34.7 ± 1.9 a	19.3 ± 0.5 d	5.3 ± 0.2 d
Kojic acid	30.0 ± 1.0 b	29.7 ± 0.5 b	20.3 ± 0.5 b
Prochloraz	9.7 ± 0.5 d	-	0.13 ± 0.01 g
Quinolinic acid	32.0 ± 1.4 ab	33.6 ± 0.5 a	46.9 ± 0.3 a
Thiophanate methyl	33.0 ± 0.9 a	23.7 ± 1.4 c	11.1 ± 0.7 c
Thiram	8.8 ± 1.1 d	-	0.52 ± 0.06 f
Control ^1^	33.3 ± 0.5 a	33.3 ± 0.5 a	-

^1^ Control experiment was performed in the absence of fungicides. ^2^
*A. alternata* was placed in the center of a Petri dish containing PDA medium and the corresponding fungicides. ^3^ Differences between means in the same column were considered significant when *p* ≤ 0.05.

**Table 5 jof-08-00547-t005:** Lesion length caused by *Alternaria alternata* and the efficacy of fungicides.

Fungicide	Lesion Length ^2^ (mm)				Efficacy (%)			
	3 Days after Inoculation	4 Days after Inoculation	5 Days after Inoculation	6 Days after Inoculation	3 Days after Inoculation	4 Days after Inoculation	5 Days after Inoculation	6 Days after Inoculation
*p*-Aminobenzoic acid	1.7 ± 0.6 c	4.0 ± 0.7 b	5.9 ± 1.8 c	9.7 ± 2.0 c	43	44	50	46
Carbendazim	2.2 ± 0.3 bc	4.0 ± 0.6 b	6.9 ± 1.5 bc	11.1 ± 1.5 bc	28	45	42	38
Difenoconazole	2.8 ± 0.8 abc	6.9 ± 0.7 a	11.6 ± 1.3 a	18.4 ± 1.5 a	8	4	2	−3
Dipicolinic acid	2.8 ± 0.1 a	6.2 ± 0.6 a	8.4 ± 1.8 abc	13.6 ± 1.8 b	7	14	29	24
Flusilazole	3.2 ± 0.5 a	6.6 ± 1.1 a	10.8 ± 1.5 a	15.4 ± 2.7 ab	−6	9	8	14
Gentamicin	2.7 ± 1.0 abc	7.2 ± 1.6 a	11.0 ± 1.7 a	16.0 ± 2.6 ab	11	13	7	11
Kojic acid	3.0 ± 0.6 ab	7.1 ± 1.0 a	11.7 ± 1.0 a	17.9 ± 0.9 a	0	2	1	0
Prochloraz	2.5 ± 0.8 abc	6.2 ± 0.6 a	9.9 ± 1.6 ab	17.6 ± 2.4 a	17	14	16	2
Quinolinic acid	2.2 ± 0.7 abc	5.0 ± 1.1 ab	8.0 ± 1.8 abc	12.3 ± 1.7 bc	26	31	32	31
Thiophanate methyl	2.7 ± 0.4 abc	7.8 ± 1.8 a	12.6 ± 2.2 a	18.6 ± 2.4 a	11	−13	−7	−4
Thiram	2.6 ± 0.6 abc	6.6 ± 1.7 a	11.1 ± 2.5 a	17.2 ± 3.1 a	13	9	6	4
Control ^1^	3.0 ± 0.4 a	7.2 ± 1.9 a	11.8 ± 2.6 a	17.9 ± 2.4 a	-	-	-	-

^1^ Control experiment was performed in the absence of fungicides. ^2^ Differences between means in the same column were considered significant when *p* ≤ 0.05.

## Data Availability

Not applicable.

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
