# Peer review of "Peel Diffusion and Antifungal Efficacy of Different Fungicides in Pear Fruit: Structure-Diffusion-Activity Relationships"

_jof, 2022, doi:10.3390/jof8050547_

Round 1

Reviewer 1 Report

This manuscript reports the evaluation of the structural characteristics that allow the diffusion of antifungal compounds in pear fruit and how the diffusion capacity influences the antifungal efficacy in vivo.  On the other hand, the paper is not suitable for publication in its present form.  Some of the revisions are listed below, and please check the manuscript. 

-Include the information in the introduction about why you choose Alternaria alternata and its importance on pears. 

-Include the definition and calculation method of EC50 value in the material and methods section.

-In Table 5, statistical results were given. Please clarify and indicate whether the statistical analysis is based on all values (fungicides and time periods) or on a day-to-day basis. 

- Discussion should be clear and concise. But it is not easily understandable and readable.  Please reorganize and rewrite the discussion. 

 There are also grammatical errors and unclear sentences in the manuscript, and it should be edited for language.

Author Response

We want to thank to the reviewer for the comments and time. All indications have been followed and addressed in the manuscript. For better understanding, a version of the manuscript highlighting all modifications in red color has been uploaded in the resubmission.

Reviewer 1

This manuscript reports the evaluation of the structural characteristics that allow the diffusion of antifungal compounds in pear fruit and how the diffusion capacity influences the antifungal efficacy in vivo. On the other hand, the paper is not suitable for publication in its present form. Some of the revisions are listed below, and please check the manuscript. 

-Include the information in the introduction about why you choose Alternaria alternata and its importance on pears. 

Answer: A new paragraph was added in lines 38-44 discussing the relevance of Alternaria alternata on pears. The ability of A. alternata to colonize pear skin and invade the internal parts of the fruit was highlighted in lines 48-50. New references were added in the manuscript (references 7-10 and 13-15). The reason for using A. alternata was added in lines 83-85.

- Include the definition and calculation method of EC50 value in the material and methods section.

Answer: Definition and calculation method were indicated in lines 156-158. A new reference (reference 37) was added to support the employed method.

- In Table 5, statistical results were given. Please clarify and indicate whether the statistical analysis is based on all values (fungicides and time periods) or on a day-to-day basis. 

Answer: To clarify this issue, “Differences between means in the same column were considered significant when P ≤ 0.05” was added as an annotation in Table 5.

- Discussion should be clear and concise. But it is not easily understandable and readable. Please reorganize and rewrite the discussion.

Answer: For better understanding, “Results” and “Discussion” sections were fused into a single section that was renamed as “Results and Discussion”. The discussions were adapted to this new format in lines 182-196, 238-257, 267-275, 286-295, 313-329, 347-353, and 366-380. A conclusion section was added in lines 382-389.

 There are also grammatical errors and unclear sentences in the manuscript, and it should be edited for language.

Answer: The manuscript was revised by a native English speaker. Several corrections were added. For example, “limiting” was added in line 14, the sentence in lines 24-25 was rewritten (“there was no relationship between stability and in vivo efficacy”), “Pyrus” was written in italics in line 34, the sentence was rewritten in lines 35-36 (“Despite its high nutritional and economic relevance”), “as” was replaced by “at” in line 100, “at” was replaced by “and” in line 113, and “larger” was replaced by “large” in lines 202. Other modifications were highlighted in red color in the revised manuscript.

Reviewer 2 Report

The manuscript entitled “Peel Diffusion and Antifungal Efficacy of Different Fungicides in Pear Fruit: Structure-Diffusion-Activity Relationships” is investigating the effects of the fungicides with various the chemical structures and diffusion capacities on the in vivo antifungal efficacy. Overall, this manuscript is well-written although some points are suggested to be clarified.

Major

  1. The data for linear regression analysis in section 3.1 and 3.2 are encouraged to be presented.
  2. Please discuss fungicide degradation in the stability tests.
  3. In section 3.3, are those EC50 relate to previous studies?

Minor

  1. EC50 should be defined.

Author Response

We want to thank to the reviewer for the comments and time. All indications have been followed and addressed in the manuscript. For better understanding, a version of the manuscript highlighting all modifications in red color has been uploaded in the resubmission.

Reviewer 2

The manuscript entitled “Peel Diffusion and Antifungal Efficacy of Different Fungicides in Pear Fruit: Structure-Diffusion-Activity Relationships” is investigating the effects of the fungicides with various the chemical structures and diffusion capacities on the in vivo antifungal efficacy. Overall, this manuscript is well-written although some points are suggested to be clarified.

 Major

  1. The data for linear regression analysis in section 3.1 and 3.2 are encouraged to be presented.

Answer: Following the indication, the data for the linear regression analysis between volume, polarity and diffusion was added in Figure 2A, the data for the linear regression analysis between volume, polarity and stability was added in Supplementary Figure S3, and the data for the linear regression analysis between diffusion and in vivo antifungal activity was added in Figure 2B.

  1. Please discuss fungicide degradation in the stability tests.

Answer: For better understanding, the “Results” and “Discussion” sections were fused into a single section named as “Results and Discussion”. Following the indication, new discussions regarding the stability results were added in lines 267-275 and 286-295.

  1. In section 3.3, are those EC50 relate to previous studies?

Answer: A new discussion was added in lines 313-329 comparing the obtained EC50 values with those previously reported. New references were added in the manuscript (references 57-59 and 62).

 Minor

  1. EC50 should be defined.

Answer: Definition and calculation method were indicated in lines 156-158. A new reference (reference 37) was added to support the employed method.